# Simplified Indoor Localization Using Bluetooth Beacons and Received Signal Strength Fingerprinting with Smartwatch

**DOI:** 10.3390/s24072088

**Published:** 2024-03-25

**Authors:** Leana Bouse, Scott A. King, Tianxing Chu

**Affiliations:** 1Department of Computer Science, Texas A&M University-Corpus Christi, 6300 Ocean Drive, Corpus Christi, TX 78412, USA; scott.king@tamucc.edu (S.A.K.); tianxing.chu@tamucc.edu (T.C.); 2Innovation in Computing Research, Texas A&M University-Corpus Christi, 6300 Ocean Drive, Corpus Christi, TX 78412, USA; 3Conrad Blucher Institute for Surveying and Science, Texas A&M University-Corpus Christi, 6300 Ocean Drive, Corpus Christi, TX 78412, USA

**Keywords:** indoor tracking, Bluetooth Low Energy, smartwatch, mobile tracking, indoor positioning systems

## Abstract

Variations in Global Positioning Systems (GPSs) have been used for tracking users’ locations. However, when location tracking is needed for an indoor space, such as a house or building, then an alternative means of precise position tracking may be required because GPS signals can be severely attenuated or completely blocked. In our approach to indoor positioning, we developed an indoor localization system that minimizes the amount of effort and cost needed by the end user to put the system to use. This indoor localization system detects the user’s room-level location within a house or indoor space in which the system has been installed. We combine the use of Bluetooth Low Energy beacons and a smartwatch Bluetooth scanner to determine which room the user is located in. Our system has been developed specifically to create a low-complexity localization system using the Nearest Neighbor algorithm and a moving average filter to improve results. We evaluated our system across a household under two different operating conditions: first, using three rooms in the house, and then using five rooms. The system was able to achieve an overall accuracy of 85.9% when testing in three rooms and 92.106% across five rooms. Accuracy also varied by region, with most of the regions performing above 96% accuracy, and most false-positive incidents occurring within transitory areas between regions. By reducing the amount of processing used by our approach, the end-user is able to use other applications and services on the smartwatch concurrently.

## 1. Introduction

There are many applications in our daily lives which rely on tracking a user’s location through mobile technology, such as car navigation, fitness tracking, and even emergency services. It is estimated that by the year 2050, 25% of the population will be aged 65 or older [1]. Older adults prefer to remain in their own home as they age [2], but certain health conditions can cause this to be potentially hazardous. ambient assisted living (AAL) technology can allow older adults to safely stay in their homes as they age, and indoor positioning is one area of AAL which can help during emergency situations [3]. If an older adult experiences a fall in their home, reliable indoor positioning could aid in the individual receiving help sooner.

Variations in Global Positioning Systems (GPSs) have been used for tracking users’ locations. However, when location tracking is needed for an indoor space, such as a house or building, then an alternative means of precise position tracking may be required because GPS signals can be severely attenuated or completely blocked. Indoor positioning systems have many approaches to solving this problem. However, many of these approaches require an extensive and time-consuming setup that would be difficult and expensive for the average end user of ambient assisted living technology. Having many calibration points, as many other approaches do [4,5,6,7,8], is one of the factors which makes it time-consuming, particularly if the system requires re-calibration. Approaches that use a form of received signal strength indicator (RSSI) need to be re-calibrated occasionally, as signal broadcasting power levels can drift and change over time [9].

As is the case for many existing solutions for indoor tracking, the initial installation, setup, and configuration of the system can potentially be very time-consuming and complicated for an end user who has only moderate experience with the pertinent technologies. Many of the current approaches to indoor localization utilize smartphones or other devices for tracking the position and movement of the user. Our approach is designed to be less computationally intensive while still achieving a competitive level of accuracy in performance on a smartwatch.

In our approach, we developed an indoor localization system that minimizes the amount of effort needed by the end user to put the system to use. This indoor localization system detects the user’s room-level location within a house or indoor space in which the system has been installed. Our primary goal is to find a solution which reduces power consumption and reduces the complexity so that this app can be used in conjunction with other AAL apps on the same device.

The developed indoor localization system uses Bluetooth Low Energy (BLE) beacons that were placed throughout a house. An Android-based smartwatch was worn by the user to scan and receive ambient BLE signals. Our approach focuses on detecting the user’s location at the region level as opposed to detecting the specific coordinate of the user within a house. This approach is useful when it is only needed to know in which room of a house the user is located. To accomplish this, we use a fingerprint-based method and the Nearest Neighbor algorithm to determine which region the user is most likely located in within the test environment.

Our contributions to the field of indoor localization systems include (1) a system with a low number of reference points and access points, (2) a system with a low-complexity algorithm that maintains an accuracy rate above 90%, and (3) a system which requires a low amount of time and effort for a user to set up and maintain. Our system has been developed specifically to create a low-complexity localization system using the Nearest Neighbor algorithm and a moving average filter to improve results. By reducing the amount of processing used by our approach, the end user is allowed to use other applications and services on the smartwatch concurrently. We have focused on reducing the amount of hardware and time-consuming steps needed for the initial setup of our system, which creates an easier overall experience for a potential end user.

The structure of this paper is divided into six sections: introduction, literature review, methods, evaluation, results, and conclusions. (1) We introduce the subject and our approach to the issue. (2) We review the current literature on the common approaches for indoor tracking and localization. (3) We move on to the methods of how we developed our system. We also discuss the equipment used, how the hardware is set up, and the different stages of operation for the system. (4) We detail our evaluation of the system, including the environment, how the data are collected, and the analysis technique used. (5) We review the results of our evaluation, and (6) our paper concludes with the conclusions section.

## 2. Literature Review

The literature reveals that many approaches for indoor tracking using Bluetooth have similar experimental environments. A common setting for experiments is a computer laboratory or similar type of room with the only obstacles being tables, chairs, and computers [10,11,12]. These laboratories were typically used by not only those conducting the experiment but also other individuals in the same communal workspace. There were also electronic devices using Wi-Fi, Bluetooth, and other wireless communications that could potentially cause interference in the experiments’ Bluetooth Low Energy beacons in the same workspace. Some experiments were spread across multiple rooms and the adjoining corridors, with beacons distributed evenly throughout the space [4,11,13,14]. One particular experiment was conducted along a single, long corridor, with the Bluetooth beacons distributed evenly along it [15]. The locations described in these approaches combined a mixture of controlling the environment in terms of exact placement and layout and including enough uncontrolled factors to simulate a realistic environment. Some of these uncontrolled factors included individuals working in the environment on other projects, active electronics unrelated to the performed research, and potential interference from these devices’ wireless communications. One approach, however, deviated from this typical experimental setup by taking place in a custom-made “Smart Lab” [16]. The lab was modeled after a 25-square-meter efficiency apartment. It was divided into a bedroom, kitchen, work area, and living room. The different areas were separated by walls that almost reached the ceiling.

A paper on activity recognition used Radio Frequency Identification (RFID) to localize where the user was in relation to several objects in an area [17]. Although the objective of this work was to determine what activity a user was performing, the approach is very similar to many indoor localization techniques. This work used RFID tags (to localize several objects in an area) and four antennae surrounding the area. The RSSI (Received Signal Strength Indicator) from each antenna was used to calculate the location of each different object. Additionally, the individual in the testing scenarios had an RFID-tagged bracelet on each wrist. The proximity of the wrist tags to the object tags was used to determine what activities were being executed by the user.

Another approach for activity monitoring which could also be used for indoor localization utilized infrared sensors. In one approach, a house was equipped with multiple infrared sensors in various locations in the house [18]. The focus of the sensors was not to interpret what actions were being performed by residents but to monitor the level of activity in areas of the house. These sensors were connected through a home area network to which they uploaded their sensor readings at set times during the day.

Radio Frequency Identification tags have also been implemented for fall detection in ways that may have potential for indoor localization. One approach used carpet embedded with sheets of RFID tags to create a binary image of objects above the carpet using the RSSIs between the tags in the carpet and RFID readers placed near the ceiling [19].

In our approach, we use the Nearest Neighbor method to determine what set of pre-established RSSI readings are most similar to the live RSSI readings collected by the watch. One approach utilized the Euclidean Distance Correction algorithm for an indoor tracking system using Bluetooth [20]. Their approach shows to be beneficial in comparison to only using the original Euclidean distance formula. Another work that uses Bluetooth for indoor tracking focused on using a method called fingerprinting to determine the distance [14]. Sample windows are an important consideration in working with fingerprinting and moving average filters as the received signal strength can fluctuate drastically. In addition to reading fluctuation, a different study explored the range and transmit power of Bluetooth Low Energy beacons [11]. Although the focus of this research was not entirely on indoor tracking, they provided valuable information regarding how objects and obstacles can impact signal strength and range. Other approaches for improving how distance is determined using Bluetooth received signal strength used the Bluetooth signal propagation model with Kalman filtering to accommodate for RSSI signal drift [9].

An approach which works very similarly to our work [21], reverses the Bluetooth receiver and Bluetooth scanner. This approach has Raspberry Pi beacons which receive the RSSI from a specific user’s device, filtering using the MAC address of the user’s device. They were able to achieve up to 91% accuracy with their approach. These researchers also went on to include this approach in a plug load management system, called Plug-Mate [22]. They used their indoor tracking approach to determine which rooms or areas are not being used and to reduce the power consumption when it is empty.

Approaches which use Wireless Sensor Networks (WSNs) in place of Bluetooth can also use RSSI analysis to achieve indoor localization. One work used the RSSI from a WSN for their localization [23]. They used a Support Vector Regression (SVR) approach, as well as an SVR+Kalman filter (KF) approach, and were able to demonstrate in a simulated environment that it preforms better than a trilateration approach.

An approach for occupancy detection in emergency situations had a similar method to our work in localizing occupants; however, the processing was performed remotely [3]. A remote control server received the RSSI values and beacon IDs through the internet and determined through a support vector machine (SVM) model what area the user was located in.

Some of the approaches used the addition of Inertial Measurement Units (IMUs) and the context of the area to increase the accuracy of RSSI-based algorithms [5,24,25]. These approaches used the IMUs to determine how far the user was moving and how likely it was that they were able to reach the RSSI-predicted location in the amount of time passed. For example, if the RSSI-based localization algorithm determined that the previous location and the current location were on opposite sides of the house, the IMU would be able to determine how likely this is to be true. Many of the researchers show an improvement in their approaches.

A similar approach is using spatio-temporal analysis of the user’s recent activity and the understanding of surrounding Points of Interest, such as the Next POI recommendation approach [26], which uses a unified neural network framework. Although this approach is not confined to indoor tracking, the concept could be applied to it.

Another work on POI-based localization for vehicles had a similar approach [27]. The paper used a gradient-based model for POI prediction for commercial vehicles based on GPS data and validation by the vehicle drivers during the preliminary stages of the research.

There are also approaches to indoor tracking and indoor localization that use Wi-Fi instead of Bluetooth [5,6,7,24,28,29]; however, the approach is very similar and involves comparing the RSSIs of Wi-Fi broadcasters from different positions in an area. One of the issues that can be encountered with these approaches is that reading Wi-Fi data consumes more battery usage when compared to Bluetooth [30].

All approaches include calibration for many reference and/or access points before the tracking period can be conducted. This type of approach can make it difficult in the initial setup as well as any re-calibration that might be needed. Our goal is a system that is more accessible for elderly people to use, and reducing the complexity or effort/time needed can make it easier on the end user. The less external assistance a user needs in managing the system, the more accessible it is.

In a majority of papers surveyed, data from reference points were collected while the user was stationary. Although collecting RSSI data while moving around the room, or from multiple RPs in a region might produce better results, our goal was to develop an approach which included minimal setup.

## 3. Methods

The indoor localization in our approach focuses on determining which room or region of the study area (referred to as the “house” or “home”) the user is in, as opposed to determining the exact position of the user. The testing area, or house, is divided into areas which we refer to as regions. Each region contains one BLE beacon, which acts as an access point (AP). Each region has one reference point (RP), which is used for creating the fingerprint for that region.

The house is separated into regions in which one beacon is placed. An example of a region is a room, such as a kitchen, living room, or bedroom. Each region has an associated region profile, which is used in establishing expected RSSI values for that region. The user is equipped with an application on a wearable smart device, which continuously receives the RSSIs from all beacons within range. The system uses this information to determine in which region the user is most likely positioned.

### 3.1. Equipment

Our research was designed to require as little hardware as possible to minimize costs and installation efforts in a home. As such, there are two main types of equipment used in our approach: a commercially available smartwatch and multiple Raspberry Pi devices.

The device we used for our smartwatch was the ASUS ZenWatch 2 [31] by ASUSTeK Computer Inc. (Taipei, Taiwan), which supports Bluetooth Low Energy. It runs the Android operating system and uses an Android application which we developed to allow the smartwatch to act as the BLE scanner. Alternatively, this system can also work on any Android-supported smartwatch, regardless of manufacturer. The only hardware requirement is that it contain Bluetooth Low Energy. With minor changes to the Android application, this approach would also work on a smart phone or any other Android device with Bluetooth Low Energy.

The Raspberry Pi in our work is used as the Bluetooth Low Energy (BLE) beacon. It is essentially a small computer that is approximately 85 mm × 56 mm × 17 mm in size, not including external casings. The model we use is the Raspberry Pi 3 B+ [32]. The primary component which we are interested in is the support for Bluetooth Low Energy. The BLE beacon has an approximate range of 23 m in an indoor location [11] and there is no proprietary software needed for our beacon. We use one Raspberry Pi per region in our experiments.

The purpose of these beacons is to broadcast an advertising packet over Bluetooth, which will be detected by a custom app running on the smartwatch. Each beacon broadcasts its device address along with its advertising packet, which is used by the app to differentiate between each beacon. The app stores information regarding all beacons in the house, which we refer to as the Beacon Network.

### 3.2. Regions

An important decision is determining how many beacons are required in the house and how they should be placed. Our current approach is to allow for one BLE beacon per region. Although the Raspberry Pi 3 B+ has an average range of 23 m, for best signal strength a beacon should be placed at least every 5–15 m [11]. The range of the BLE beacon decreases significantly when there are large obstructions, such as walls or large furniture, between the beacon and the BLE scanner. For our purposes, where regions are typically defined as a room in a house, this can be a benefit. This can allow for more unique sets of Bluetooth signal readings gathered by the smartwatch. However, obstructions within a region must still be taken into consideration. For example, if a beacon is placed along the wall next to a corner and the BLE scanner is on the other side of the corner, then there is significantly reduced reception. This may impact the accuracy of the indoor localization system. If a room is not larger than 5–15 m but contains corners, then additional beacons may be needed. With these concerns in mind, for our approach, we place beacons so that they are at the maximum distance from each other.

Each region has a corresponding region profile, which consists of the average readings of each beacon within range. These readings are initially gathered during a calibration phase. Each region’s profile is unique. For instance, the average RSSI for beacon ‘A’ will be different when the user is standing in the kitchen compared to standing in the bedroom.

### 3.3. Hardware Setup

The setup of the hardware for our system is a simple process. The Raspberry Pi devices that we use as beacons are labeled with a beacon number with which each is associated. The user simply plugs the beacon into a power outlet in the region for that beacon. For example, Beacon 1 would be placed in Region 1. Preferably the beacon would be placed in a position farthest from other beacons. This allows more variations in RSSIs from region to region. The user then runs a file on the Raspberry Pi device which ensures the Bluetooth Low Energy broadcasts an advertising packet which contains the beacon’s MAC address.

Although there is a preference for placing the beacon farthest from other beacons, for example, in the farthest corner of the room, there is no requirement for it to be placed in a specific place. Provided there are no significant obstructions to block the beacon, the beacon can be placed where it is convenient for the user.

Once this is completed, the user then opens the indoor tracking application on their smartwatch, and they are ready to begin the calibration and tracking phases.

### 3.4. Offline Phase

The offline phase of our system consists of calibrating the beacon profiles, which are used in the online phase to determine the location of the user. The calibration phase records the standard Bluetooth Signal Strength of all beacons from each room. From these recordings, we construct the fingerprints from each beacon.

#### 3.4.1. Fingerprinting

The indoor localization is based on the fingerprinting method using the Received Signal Strength (RSS) of the Bluetooth LE beacons within range. This is a common method used for indoor positioning systems and works for multiple types of signals, such as Bluetooth, Wi-Fi, and Radio Frequency Identification (RFID). Fingerprinting essentially compares the current RSSIs of all beacons within range to sets of known RSSI values throughout a specified area [14]. Though this method can be used for either Wi-Fi or Bluetooth localization, we will be using Bluetooth Low Energy to reduce battery consumption [30].

This approach operates on a simple overall concept: there are multiple signal broadcasters placed throughout a house, ideally at least one to each room. In our case, we use Bluetooth Low Energy beacons as our signal broadcasters. Each beacon has a unique identification (ID) number. When the user stands in each region of the house, they wear a device which acts as a BLE receiver. For each location in the house, the signal strength from each of the beacons will be different. Figure 1 illustrates an example of the readings collected from a specific beacon while in different regions. We can see in each of the regions that there is a distinctive difference in RSSI ranges for the same beacon, varying depending on where the user is in the home. We finally compare the current signal strengths of all the beacons to a list of known signal strengths recorded in different positions throughout the house for localization purposes.

Fingerprinting can be broken down into two primary parts: access points and reference points.

Access points (APs) are the stationary points which are broadcasting the signal that is received by the user’s device. In our work, the Bluetooth beacons act as our access points. Our beacons continuously broadcast a Bluetooth Low Energy advertising packet to all devices receiving Bluetooth packets within range. The packet contains the MAC address of the beacon that is broadcasting. The smart device has a record of all the beacons in the beacon system, along with the MAC address for each.

Reference points (RPs) are the known positions throughout the testing area where the RSSI values from all of the APs within range in the beacon system are recorded. The collection of RSSI values for a reference point is referred to as the fingerprint. The RP’s fingerprint is calculated during the calibration phase. In our work, each region contains one reference point and one access point. The reference point we use is typically in the center of the region. During the calibration process, the user will stand in place for 180 s. Then, the user is prompted by the application to relocate to the next region. This calibration process is repeated for each region in the system.

In addition to focusing on the algorithms and hardware used for this research, reducing the number of access points or reference points is also a concern. Although there have been approaches with less than 50 reference points [6,7], there have been approaches with over 100 [4,8], or even over 2000 [5] reference points. Because our approach is focused on room-based or region-based localization, we are able to utilize only one AP and one RP per region in the household.

#### 3.4.2. Calibration

Each of the beacons continuously broadcasts a Bluetooth LE advertising packet that contains the beacon’s MAC address. The smartwatch application has a list stored of all beacons, with their ID number, MAC address, standard assigned name (“Living Room”, “Kitchen”, etc.), and the region to which they belong. Each region has a calibration profile with which it is associated. The MAC address, beacon number, and standard assigned name are currently hard-coded into the smartwatch application for the development and evaluation stage of our research. However, this information can easily be set with user input in the application.

The calibration process begins when the user activates the calibration mode on the smartwatch. The user is instructed to move to the center of the first region and to press a button indicating they are ready to begin. The smartwatch application collects and stores the RSSI readings of all beacons within range for a duration of 180 s. Once the calibration has been completed for that region, the user is prompted to move on to the next region and repeat the same process.

Once the process has been completed for each region, the profile is calculated. For each region, the collection of RSSI readings is grouped by beacon ID in ascending order by their timestamps. Then, each beacon ID group is used to gather data for the fingerprint. These data consist of the mean, variance, standard deviation, and average timestamp for all the readings. Algorithms 1 and 2 demonstrate this calibration process.

In our approach, each region profile has one fingerprint for every beacon in the system. The fingerprint consists of the mean and standard deviation of the collection of RSSI values for that beacon ID grouping. An example of the profile for Region 0 can be seen in Table 1.

Once the offline phase calibration has been completed, the region profiles are saved to the device.
**Algorithm 1** Region calibration1:**procedure** Calibrate(Boolean filter, DataFrame RSSI)2:        **for** each region_id *i* **do**3:              **for** each beacon_id *j* **do**4:                    **if** filter==TRUE **then**5:                        RSSI[i][j] = MovingAvg(RSSI, 2)6:                    profile[i][j] = getFingerprint(i, j, RSSI[i][j])7:        **return** profile

**Algorithm 2** Calculate fingerprint for a set of RSSI values of a beacon
1:**procedure** getFingerprint(Int region_id, Int beacon_id, DataFrame RSSI)2:      fingerprint[‘mean’] = RSSI[‘RSSI’].mean()3:      fingerprint[‘timestamp’] = RSSI[‘timestamp’].mean()      ▹ Average the timestamps4:      **if** region_id==beacon_id **then**5:            fingerprint[‘weight’] = 0.756:      **else**7:            fingerprint[‘weight’] = 1.08:      **return** fingerprint


### 3.5. Online Phase

After all of the beacons have been calibrated, the user can then enter the tracking phase. In this phase, the indoor localization app scans for all of the registered BLE beacons and reads their RSSI values. The app then takes these readings and compares them to each beacon’s profile. Whichever profile is closest to the current readings will determine which beacon the user is closest to—and therefore in which room or area of the house the user is located.

#### 3.5.1. RSSI Sampling

There are a few steps to take to determine which beacon is closest to the user and therefore in which area of the house they are located. The first step is to determine the width of the RSSI sampling window. When the BLE scanner reads the RSSIs from the beacons, regardless of whether the user is moving, the strength of the signal does not remain steady. The signal fluctuates in strength but does show a clear trend in values depending on the distance between the user’s smartwatch and the beacon. However, if only a single RSSI reading is considered it could very easily produce the wrong result. A window size that is too short can result in not enough data. Likewise, a window that is too long can result in possible “smudging”, which is common if a user is walking from one room to another, or even between more than two rooms, in the span of one sample window. In this case, smudging would be when readings from two or more rooms are collected in the same sample window for averaging. This can result in inaccurate location predictions.

#### 3.5.2. Tracking

During the tracking phase, the localization application gathers the current RSSI readings and compares them to the fingerprints for all of the regions to determine which is most similar. Because Bluetooth signals can fluctuate, we take an average of readings over 10 s of all beacons in range [33]. For our evaluation, we also compare how effective our approach is using durations of 3 and 5 s for averages. During this phase, we use this moving window average instead of a Gaussian filter to reduce power consumption since these readings are taken continuously. We compare the readings against each set of fingerprints and assign it a score based on the weight of the fingerprint and how far away it is from the fingerprint.

When comparing the current readings to each set of fingerprints, there are different weights to each beacon in a fingerprint. The beacon which belongs to the corresponding region has a weight of 0.75, while the rest of the beacons in that fingerprint have a weight of 1.0. For example, in Figure 2, the weights for Region 0 and Region 1 would be as they are in Table 2, with Beacon 0 in Region 0 having a weight of 0.75, Beacon 1 in Region 1 having a weight of 0.75, and all other beacons in these regions having a weight of 1.0. The weight of 0.75 allows the algorithm to place a priority on that region’s beacon. This number was determined through a series of trials testing different values. Both lower and higher values decreased the accuracy, while 0.75 increased the accuracy.

To determine which region the user is located in, we utilize a weighted Nearest Neighbor algorithm, seen in Algorithm 3, and we determine what the weighted score is for each region’s profile. First, we take the current average of RSSI readings from each beacon within range. This is considered the fingerprint of readings. We then compare the current fingerprint RSSI readings to each beacon’s fingerprint profile. The difference is subtracted from each RSSI pair—that is, the current RSSI reading for Beacon A is subtracted from Region 1’s reading for Beacon A, and then the current reading for Beacon B is subtracted from Region 1’s Beacon B, and so forth until we have the difference in values for each beacon for that region. The differences for each beacon in the region are then multiplied by the appropriate weight, as described above. Finally, all the values for that region are combined as the weighted score for that region. Algorithm 4 demonstrates a simplified version of the overall process.

This process is repeated for each region profile. Whichever region has the lowest weighted score is determined to be the region the user is closest to.
**Algorithm 3** Determine the Nearest Neighbor  1:**procedure** NearestNeighbor(profile, fp)  2:      **Group** profile by region_id  3:      **for** each region_id group i **do**  4:            *w* = 0, *t* = 0  5:            **for** j in (length(group[i]) **do**  6:                  t = pow((fp[j][‘mean’] − group[i][j][‘mean’]), 2)  7:                  t = t × group[i][j][‘weight’]  8:                  *w* += *t*  9:            wts[*i*] = [region, group[i][j][‘timestamp’].mean(), *w*]10:      **return** wts.minimum()                                    ▹ Return region with lowest weight

**Algorithm 4** Simulated tracking
  1:**procedure** Tracking(profile, RSSI, filter, freq)  2:      **Group** all readings by frequency                                 ▹ 3 s, 5 s, or 10 s  3:      **for** each frequency group *i* **do**  4:            **if** Group *i* not Empty **then**  5:                  **Group** RSS[i] by Beacon_id  6:                  **for** each Beacon group *j* **do**  7:                        **if** filter==TRUE **then**  8:                             RSSI = MovingAvg(RSSI, 2)  9:                        fp[j] = getCurrentFingerprint(RSSI)10:                  prediction[i][‘location’] = nearestNeighbor(fp, profile)11:                  prediction[i][‘true_loc’] = round(group[‘true_loc’].mean())12:      **return** prediction


## 4. Evaluation

To evaluate our approach, we conducted a test in a home environment. Live data were collected from hardware during the test and were then processed through several simulations of our indoor localization system using varying parameters. This allowed us to compare different variations in our approach to determine the most effective one on the same collection of data.

The simulations were separated into two primary parts, first using five regions, and the second approach using only three of the regions. Both of these approaches were derived from the same collected dataset to ensure an accurate comparison.

### 4.1. Environment

The evaluation was conducted inside one of the researchers’ residences across five rooms of the home. The layout of the home can be seen in Figure 2, which is not to scale. The overall area for the home is 7.32 m by 13.41 m, which is approximately 321.87 square meters. Each region consists of a single room, with one beacon present in each region. Each of the walls dividing the rooms is approximately 10.5 cm thick, with furniture arranged throughout each room as indicated by rectangle boxes on the diagram. All beacons were placed approximately 1 m from the ground, and the smartwatch was held at a steady 1 m from the ground throughout the experiment.

As this was a home environment, there were several Bluetooth- and Wi-Fi-connected devices in use at the time of the experiment. This aspect of the testing area helps in determining the efficiency of our system within a realistic environment.

### 4.2. Data Collection

Before tracking data were gathered for each session, we performed the calibration sequence as described in the previous section. The number of regions, number of beacons, and the beacon MAC addresses were hard-coded into the Android application and changed manually depending on what was needed for each experiment. For each room in the testing area, we gathered the RSSI readings in the center of each room for 180 s. After the 180 s had elapsed, the user was prompted to relocate to the next region and press the button when ready to begin for that region. We repeated this process in each of the regions to be included in testing for that session.

During the calibration sequence, all RSSI readings collected were recorded raw, without alteration, and saved to a file on the device.

Once the calibration data collection was complete, we moved on to the tracking phase. For each region, we designated five positions for the user to stand stationary in. We have chosen to have five stationary positions for experimenting to better understand where areas of low accuracy might be found in a home environment. Readings were collected from each position for five minutes before the user moved on to the next position in the region. The same process was repeated for all regions. This number, as well as the 180 s for calibration, was determined in consideration of the total testing time needed with combined calibration and tracking phases. Three minutes per room, with five rooms, makes fifteen minutes for the calibration phase. Then, with an additional 5 min per position in each room, the total was 125 min for tracking—making a total of 140 min for both phases of the testing. The placement of these positions can be seen in Figure 3, with each position indicated with a black circle and the label P0–P4. The Android application was set to collect RSSI values every ten seconds and save the collected data to a file. For this experiment, we appended all received RSSI data to a raw data file for analysis, including the timestamp for each reading.

We performed this experiment first using five BLE beacons across five regions and then again with three BLE beacons across three regions. The placement of beacons in the first experiment can be seen in Figure 2, with Beacons 0–4. The placement of beacons in the second test is the same as the first but only uses Beacons 0–2.

To determine the ground-truth accuracy for which readings were associated with which regions, the experiment was performed by one of the researchers of this work and recorded with a web camera. The timestamps for entering and exiting each of the regions were manually recorded in a file. Based on these timestamps, a Python script was written to insert the ground-truth location for each raw RSSI reading into the data file. Table 3 displays an example of tracking data collected during this phase of the experiment.

### 4.3. Technique Analysis

After the data have been collected, we transfer them to the computer for simulating the tracking process with different variations. We used the same tracking algorithm for each of the variations, which were (1) the application of a moving average filter, (2) RSSI sample aggregation frequency, and (3) the number of beacons. We wanted to determine the effectiveness of applying a simple moving average filter to the gathered RSSI data, which can be seen in Algorithm 5. For each of the experiments and variations, we ran the simulation with and without the filter on both the calibration data and the tracking data. In addition to this, we wanted to determine how the performance fares aggregating the RSSI samples at different frequencies—for example, determining the user’s location every 5 s. We chose three sample frequencies: 3 s, 5 s, and 10 s.

The moving average filter is an important step in our approach as it is a simple means of reducing the amount of fluctuation in the RSSI readings as they are collected by the Android application. Figure 4 displays a sample of collected RSSI data for Beacon 4 while the user is in Region 1. The top graph displays the unfiltered data, while the bottom graph displays the simple moving average filter with a window size of 2. The red horizontal line indicates the mean values.
**Algorithm 5** Moving average filter  1:**procedure** MovingAvg(List RSSI, Integer window)  2:      *x* = [0] × len(RSSI)  3:      **for** *i* < length(RSSI) **do**  4:            sum = RSSI[*i*]  5:            count = 1  6:            **if** *i* < window **then**  7:                 **for** *j* < window **do**  8:                       sum += RSSI[*i* + *j*]  9:                       count += 110:            **else if** *i* > (length(RSSI) − window) **then**11:                 **for** *j* < window **do**12:                       sum += RSSI[*i* − *j*]13:                       count += 114:            **else**15:                 **for**
*j* < window **do**16:                       sum += RSSI[*i* − *j*]17:                       sum += RSSI[*i* + *j*]18:                       count += 219:                 *x*[i] = sum/count20:      **return** *x*

The first step in our simulation is to load the raw RSSI data collected during the calibration process. We group the data by region, then by Beacon ID, and pass them through a moving average filter with a window of size 2. This size was chosen after a trial-and-error process to determine which value produced the best results. Once we have both the raw and the filtered data, we use these sets of data in our calibration algorithm, which was seen previously in Algorithm 1. The calibrated region profiles are then saved for use in the tracking portion of the simulator.

Once we have calibrated the beacons for each variation of the experiment, we are able to move on to simulating the tracking portion of our experiment. We can see the overall algorithm for determining the current location of the user in Algorithm 4, the Simulated Tracking algorithm. This algorithm loads the raw RSSI data that were collected and then groups them according to timestamps based on the frequency for aggregation of the data. We run the full simulation experiment using first a 3-s frequency, then a 5-s frequency, and finally a 10-s frequency. Then, for each of these frequencies, we run the simulation experiment first without a moving average filter and then with the moving average filter. The size of the Moving Average Window varied slightly between tests, depending on the frequency. For the ten-second aggregation frequency we used a window size of 3, while for both five- and three-second frequencies we used a window size of 2.

After the current group of data have been filtered, the current fingerprint of all the beacon readings within range is collected using Algorithm 2. This essentially determines the mean, variation, and average timestamp for the readings.

The current fingerprint of the beacons is then used to determine which region is closest to the user. We use a weighted Nearest Neighbor, seen in Algorithm 3, to determine this. This algorithm calculates the Euclidean distance between our current collection of RSSI values (the current fingerprint) and the calibrated region profile RSSI values. The typical Euclidean distance formula determines the distance between two vectors *p* and *q*. In our application, the two vectors are the fingerprint of a region’s profile and the fingerprint of the current RSSI readings for beacons within range.

We have adjusted the Nearest Neighbor algorithm to allow different weights for each region’s RSSI values. In the original distance formula, the distance between the two vectors’ values is the sum of the difference of each value raised to the power of 2. The two vectors being subtracted represent the RSSI values for each beacon from (a) the user’s current position and (b) one of the region profiles. As previously stated, each beacon in a region profile has a weight assigned to it. For each region profile, the beacon which resides inside of the same region as the profile has a weight of 0.75, while all other beacons have a weight of 1.

When distance is calculated between corresponding beacons in the two vectors, we multiply the outcome by its weight before adding it to the summation. As the region profile with the lowest difference in RSSI values to the current readings is the closest region to the user, our weighted algorithm allows more importance to be placed on beacons that correspond with the region profile being evaluated.

Once all of the weighted Euclidean distances between the current RSSI readings and each of the region profiles have been calculated, we are able to compare the values. Whichever region profile has the lowest value is determined to be the region to which the user is closest.

## 5. Results

In our experiments, we compared results using a variety of variables. We tested results using five beacons and three beacons as well as varying the amount of time spent aggregating RSSI readings for accuracy. In our evaluation, accuracy refers to whether the algorithm predicted the correct region for where the user was. We also compared the results, including a smoothing filter for RSSI readings. Experiments run with a moving average smoothing filter had the filter applied to both the calibration readings as well as the tracking readings.

We were able to achieve a 92.1% accuracy in our indoor tracking system using five beacons across five rooms, and 85.9% accuracy using three beacons across three rooms. Both of these results were attained by aggregating readings every ten seconds. Our worst results were shown with 88.9% accuracy using five beacons at five-second aggregation and 83.8% using three beacons at three-second aggregation. The full table of results can be seen in Table 4.

In addition to the overall performance, we have also looked at the performance for each individual region in each of the experiments. We can see a visual overview of some of these results in Figure 5 and Figure 6. In these heatmap plots, the color bar on the right side indicates the color associated with the number of positive readings. Diagonal cells of the heatmap are true positives (TPs) and all other cells are false positives (FPs). A true positive indicates that it correctly predicted that location to be the current location. Under the FP or TP designation is the total number of predictions for that location, followed by the percent that represents samples for that location (row). For example, in the top row, Region 4, each box on that row shows how often it predicted the user to be in Regions 0–1 while standing in region 4. At position (4,4) we can see that there was a TP rate of 96.79% accuracy. Figure 5 demonstrates our experiment run with the worst overall performance. This figure displays the number of true-positive and false-positive readings for each actual location. The overall accuracy for this test was 83.778%; however, we can see that in Region 1 there was a 93.67% true-positive rate. This discrepancy can also be seen in our best performance, with five beacons and ten-second aggregation, which can be seen in Figure 6. The overall average was 92.106%; however, we were able to achieve 98.68% accuracy in Region 3.

When we take a closer look at the breakdown of the accuracy results for each position, we begin to see a clearer picture of factors which may impact how accurate the system is in a home environment. In Region 0, there are two positions where the accuracy was 100%. This can be seen in Table 5. Meanwhile, the results for Region 3 demonstrate much higher accuracy, with three results over 95%, as seen in Table 6.

Figure 7 displays the accuracy for each position in each region of the test environment. Position 0 of Region 0 performed with the worst accuracy results and was placed in a transitory area between Region 0 and Region 1. We can see from Table 5 that for Position 0, it correctly predicted Region 0 fifteen times and incorrectly predicted Region 1 twelve times. Similarly, in the accuracy table for Region 2, Table 6, we see Position 0 incorrectly predicting Region 1, which is the adjoining room. This trend of incorrect predictions is also partially reflected in Position 3 in Region 0. Although this position is not in a transitory location, there are no walls separating it from Region 1. There is a clear path from Position 3 to Region 1 without any major obstructions, which may be causing uncertainty in the readings. In addition to this, as this evaluation took place in a home environment, there is the possibility of other wireless and Bluetooth devices interfering with the smartwatch’s ability to accurately receive RSSIs in all areas of the home.

## 6. Conclusions

Our primary goal for our research has been to develop an indoor localization system which minimizes complexity while still maintaining a high rate of accuracy in its performance. We were able to create an indoor localization system which operates with only one Bluetooth beacon per room and with only one reference point in the initial calibration process. The initial setup time can be completed in less than five minutes per room in the end user’s home.

In addition to simplifying the equipment and setup process for our system, we also reduced the amount of processing needed for detecting the user’s location by using a simple moving average filter on the processed data and the Nearest Neighbor algorithm for detecting position. We also sought to reduce battery consumption where possible, such as by utilizing Bluetooth Low Energy as our tracking method as an alternative to Wi-Fi. These energy-conserving measures may allow the user to better utilize their smartwatch for additional purposes other than solely for indoor localization.

In addition to this, because the entire process is self-contained in each smartwatch, our system can easily support multiple users on the same system of beacons with no interference. Our system overall provides an easy experience for any potential end users in its initial setup and its continued use.

There are several limitations to our work that we believe could be overcome in future works. Although our overall approach has a 92.1% accuracy rate, we believe that improvement could be made in the calibration process that would provide better results. Including multiple reference points per region, instead of only one, could provide a more diverse understanding of typical RSSI readings for each region. This would be particularly helpful in transitory regions, such as entrance ways from one region to the next. A balance between reference points, accuracy, and setup time would need to be taken into account for future works looking to improve on this aspect.

Additionally, the matter of scaling is another limitation of this approach. As it is, for a five-room living space, it would take approximately fifteen minutes to calibrate the system. Any time there is any significant change in the layout of the beacons, the system must be re-calibrated. If this system were to be deployed on a larger scale, for example, in a care home facility with significantly more rooms, then the setup time increases as well. In a large-scale application, re-calibration becomes a significant time-consuming process. Future work exploring ways to decrease this issue, without impacting accuracy, could be very beneficial.

Other areas for future work may also lie in modifications to the user interface. As the system currently is, many features are hard-coded into the smartwatch application, such as the MAC address for each beacon and its corresponding region number. Users could also be given the option to select sampling times, such as 5 or 10 s, to find a solution which works for their needs. Additionally, creating a version of the application for smartwatches would provide an alternative for those without access to or who are unable to afford a smartwatch. The only hardware requirement for this system to operate on a smartphone or tablet would be that has Bluetooth capabilities.

It is our hope that these contributions may provide members of the public with better access to and use of a user-friendly indoor localization system to ensure better quality care for themselves or their loved ones.

## Figures and Tables

**Figure 1 sensors-24-02088-f001:**
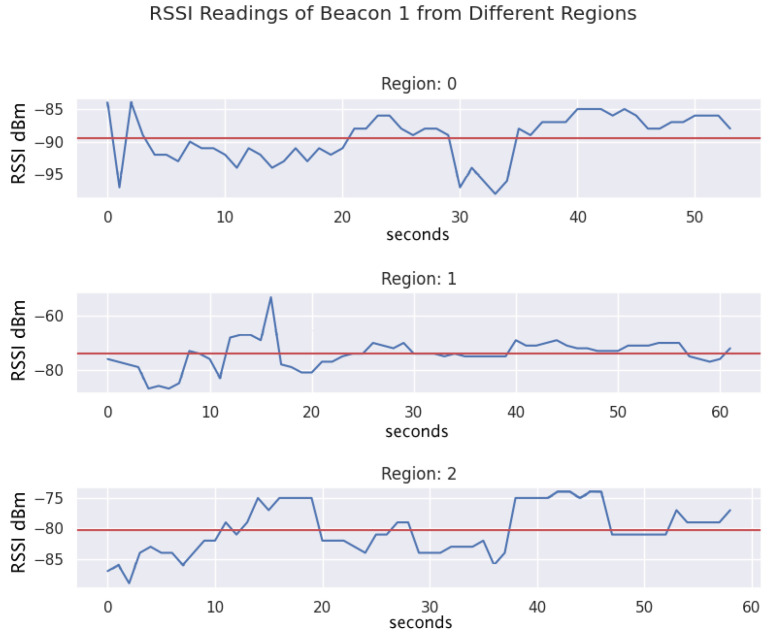
Example of RSSI readings of Beacon 1 while in different regions of the testing area. The mean is indicated as a red solid line and the RSSI signal as a fluctuating blue line.

**Figure 2 sensors-24-02088-f002:**
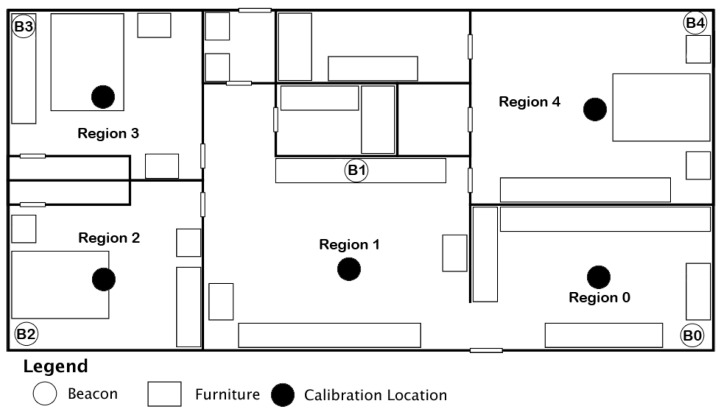
Placement of Regions 0–4, Beacons B0–B4, and calibration locations for each region in home for evaluation and data collection.

**Figure 3 sensors-24-02088-f003:**
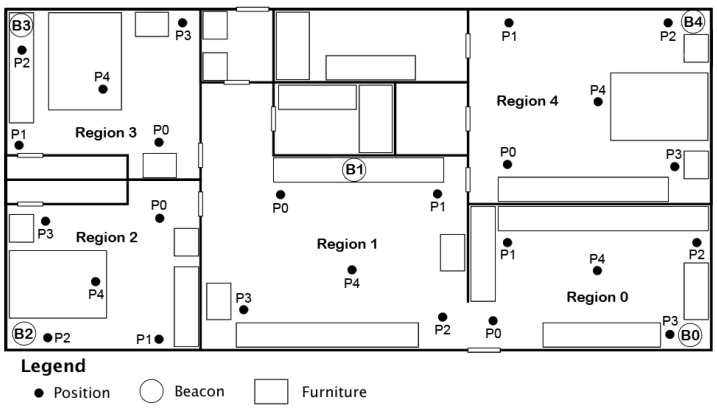
Placement of Positions P0–P4 in each region for evaluation and data collection.

**Figure 4 sensors-24-02088-f004:**
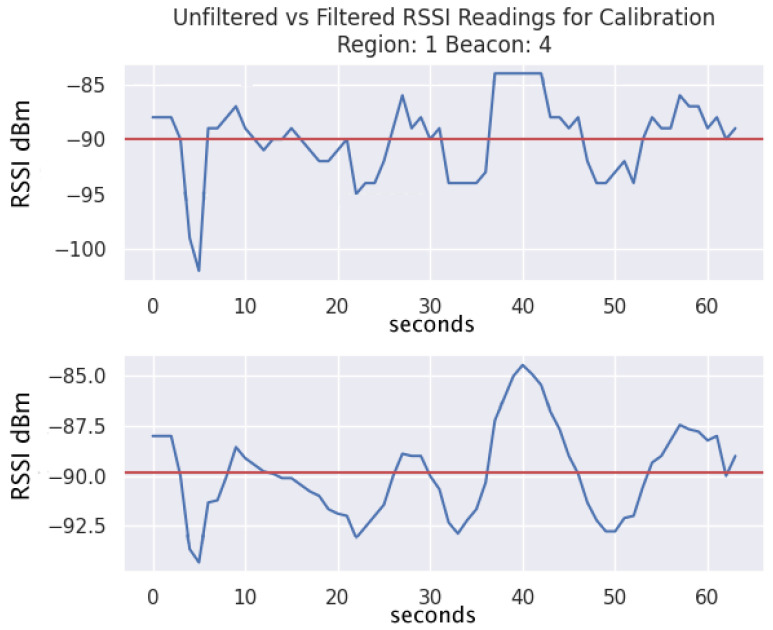
Unfiltered and filtered RSSI reading comparison, with mean indicated as a red solid line and the RSSI signal as a fluctuating blue line.

**Figure 5 sensors-24-02088-f005:**
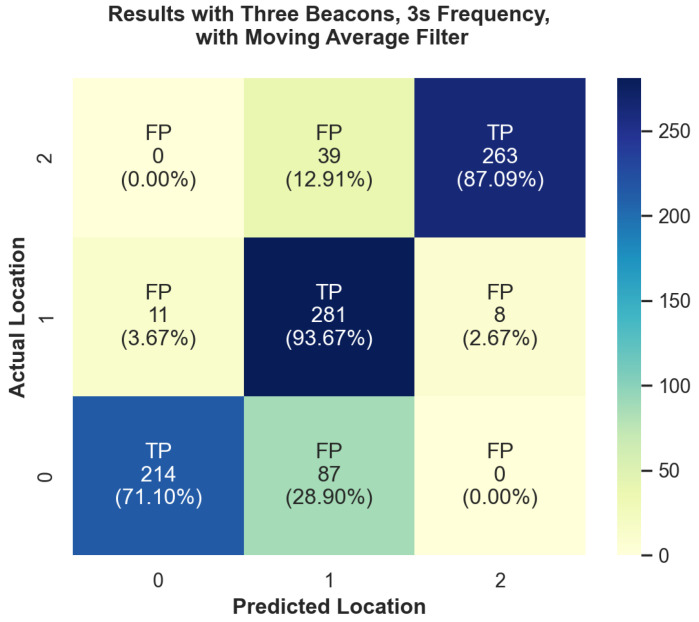
Heatmap plot of the accuracy of predicted room locations versus actual room locations for experiment with three beacons at 3-s aggregation with moving average filter.

**Figure 6 sensors-24-02088-f006:**
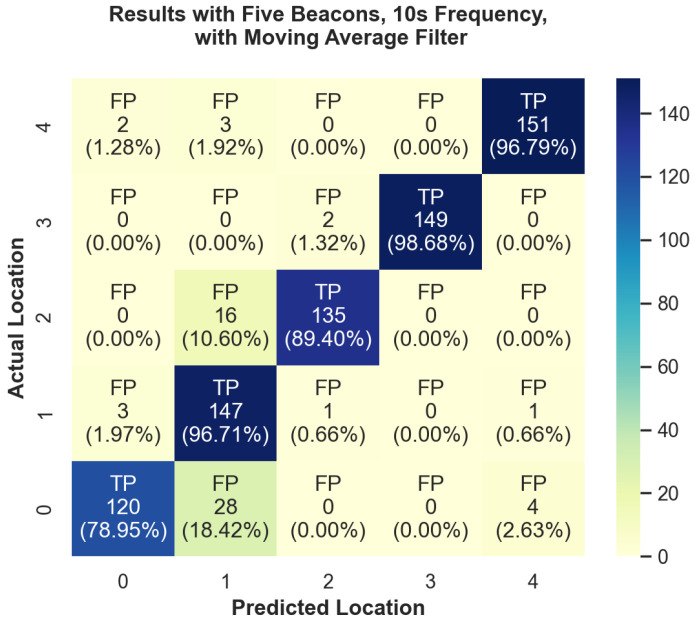
Heatmap plot of accuracy of predicted room locations versus actual room locations for experiment with five Beacons at 10-s aggregation with moving average filter.

**Figure 7 sensors-24-02088-f007:**
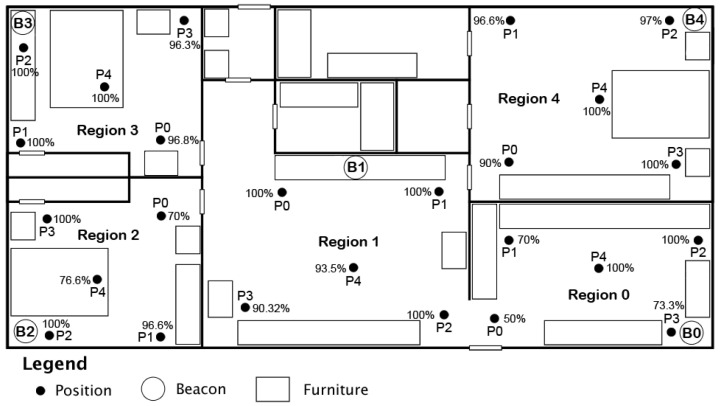
Accuracy results for each Position P0–P4 in each region.

**Table 1 sensors-24-02088-t001:** Example of stored values for calibrated profile for Region 0.

Region ID	Beacon ID	Mean	Var
0	0	−68.9613	1.6663
0	1	−85.3924	1.6692
0	2	−92.8077	1.4341
0	3	−95.0083	1.5450
0	4	−87.68789	1.7625

**Table 2 sensors-24-02088-t002:** Example of weights associated with the profile for Region 0 and Region 1.

Region ID	Beacon ID	Weight
0	0	0.75
	1	1.0
	2	1.0
	3	1.0
	4	1.0
1	0	1.0
	1	0.75
	2	1.0
	3	1.0
	4	1.0

**Table 3 sensors-24-02088-t003:** Sample entries from tracking phase of the data collection.

Timestamp	Beacon ID	MAC Address	RSSI	True Location	Position Location
2023-02-06 20:06:20	1	B8:27:EB:54:87:D0	−71	2	2
2023-02-06 20:06:20	0	B8:27:EB:D6:2F:E2	−85	2	2
2023-02-06 20:06:20	4	B8:27:EB:DB:10:50	−87	2	2
2023-02-06 20:06:20	2	B8:27:EB:2C:39:0A	−49	2	2
2023-02-06 20:06:20	3	B8:27:EB:D0:B5:91	−83	2	2

**Table 4 sensors-24-02088-t004:** Average evaluation accuracy for experiments with and without moving average filter. Best performance for 5 and 3 beacons are seen in bold.

Beacons	Filtering	10 s	5 s	3 s
5	Yes	**92.106%**	88.880%	88.973%
5	No	91.974%	88.946%	89.040%
3	Yes	**85.939%**	83.889%	83.950%
3	No	85.720%	83.778%	83.839%

**Table 5 sensors-24-02088-t005:** Accuracy table for location predictions for Region 0, listed by each position in the region, using moving average filter at 10-s frequency. The true location for this table is Region 0.

Position	Predicted Region 0	Predicted Region 1	Predicted Region 2	Predicted Region 3	Predicted Region 4	Accuracy
0	15	12	0	0	3	50.00%
1	21	8	0	0	1	70.00%
2	31	0	0	0	0	100.00%
3	22	8	0	0	0	73.33%
4	31	0	0	0	0	100.00%

**Table 6 sensors-24-02088-t006:** Accuracy table for location predictions for Region 2, listed by each position in the region, using moving average filter at 10-s frequency. The true location for this table is Region 2.

Position	Predicted Region 0	Predicted Region 1	Predicted Region 2	Predicted Region 3	Predicted Region 4	Accuracy
0	0	9	21	0	0	70.00%
1	0	1	29	0	0	96.66%
2	0	0	30	0	0	100.00%
3	0	0	31	0	0	100.00%
4	0	6	24	0	0	80.00%

## Data Availability

The data presented in this study are available on request from the corresponding author.

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
