# Peer review of "Simplified Indoor Localization Using Bluetooth Beacons and Received Signal Strength Fingerprinting with Smartwatch"

_sensors, 2024, doi:10.3390/s24072088_

Round 1
Reviewer 1 Report (Previous Reviewer 2)
Comments and Suggestions for Authors
Comments to the Author
This paper proposes an indoor localisation system that combines the use of Bluetooth Low Energy beacons and a smartwatch Bluetooth scanner to determine which room the user is located. However, there are several points that need to be addressed to improve the quality of the manuscript.
Suggestions to improve the quality of the paper are provided below:
1) It was mentioned in Section 3.5.2 that different weights are assigned to the beacons and the optimal weight of 0.75 and 1 was obtained through a series of trials testing different values. However, this seems to be a very manual process and the weights will likely change depending on the building layout. Can this process be automated in some way during the calibration process?
2) Is the approach described in Algorithm 3 robust to missing data? For instance, if the user is located in certain parts of the house that is far away from several beacons, the sensors are unlikely to capture the RSSI readings from those beacons. Will Algorithm 3 still work in this scenario? How does it address this problem?
5) In Figure 8, most of the results make sense as the points closer to the beacons and away from the region boundaries are showing higher accuracies than the points that are close to the boundaries of each region. However, there are some points that do not follow this trend. For instance, 1) P3 for Region 0 with a low accuracy of 73.3% when it is next to Beacon B0, 2) P2 in Region 1 with a high accuracy of 100% even when it is close to Region 0, and 3) P4 in Region 2 with an accuracy of 76.6% even though it is in the middle of the room. Please kindly provide an explanation for these results.
6) Please include a discussion section on the limitations of the existing approach and elaborate on how they can be improved in future works. Some discussion points include
- Depending on the adoption rate in different countries, it might not be very common that every individual has a smartwatch. In this scenario, does this approach work with the individual’s smartphone?
- The entire duration it takes for the system to be up and running seem to be much longer than the 15 mins needed for calibration as there are many system parameters that need to be calibrated to maximise the system's localisation performance. This includes the weights of the beacons, the window length for calculating the moving window average, etc. How will this be addressed in the future?
Comments on the Quality of English LanguageThere are no major issues related to the manuscript's quality of English, except for some minor issues highlighted in my current set of comments.
Author Response
Thank you for your review report on this paper. Please see the attachment for a point-by-point response to comments.

Reviewer 2 Report (Previous Reviewer 1)
Comments and Suggestions for Authors
We recommend to accept the current version of manuscript since all required modifications are done.
Author Response
Thank you for your review report on this paper. Please see the attachment.

Round 2
Reviewer 1 Report (Previous Reviewer 2)
Comments and Suggestions for Authors
Thank you for taking the time to address my comments thoroughly and comprehensively. I believe all my comments have been adequately addressed, and the quality of the manuscript has increased significantly as a result. I have determined that the manuscript is now ready for publication.
Comments on the Quality of English LanguageThere are no major issues related to the manuscript's quality of English, except for some minor issues that do not affect the clarity and flow of the manuscript.
This manuscript is a resubmission of an earlier submission. The following is a list of the peer review reports and author responses from that submission.
Round 1
Reviewer 1 Report
Comments and Suggestions for Authors
The research field of this paper, whether based on Bluetooth or Wi-Fi,
already has many similar papers. Here are some of comments that we
hope the authors to include them in the revised manuscript:
1. The reference section can include more citations about indoor positioning
techniques using other signal sources; e.g., geomagnetic.
2. For Bluetooth-based positioning technology, there are already commercial
positioning technologies, such as iBeacon. The positioning accuracy of the
method proposed in this paper seems not to be superior than these commercial indoor positioning technologies. Please explain why the authors do not use these commercial Bluetooth indoor positioning technologies directly.
3. The use of AI / machine learning methods to analyze the fingerprints of Wi-Fi signals or Bluetooth signals has been regarded as a prominent method for indoor positioning technology in recent years. However, it is obvious that the authors of this paper still use hard computing algorithms to perform Bluetooth signal fingerprints analysis and positioning. Please explain why AI method is not used to assist Bluetooth indoor positioning.
4. When detecting and analyzing the fingerprint of the Bluetooth signal, the authors only uses RSSI as a single metric to evaluate the quality of the Bluetooth signal, but RSSI may not be the only metric to measure the quality of the Bluetooth signal. Please explain why RSSI is the only metric to be used.
5. Please explain if a different model of smart watch is selected (in fact, even the same model of smart watch may not have the same characteristics of receiving Bluetooth signals), what additional calibration procedures will be required to activate the indoor positioning system ?
Reviewer 2 Report
Comments and Suggestions for Authors
Comments to the Author
This paper proposes an indoor positioning system using Bluetooth low energy beacons and a smart watch Bluetooth scanner to determine the users’ room location. Indoor localization is an important topic with many potential application areas. However, there are several points that need to be addressed to improve the quality of the manuscript.
Suggestions to improve the quality of the paper are provided below:
1) In the Introduction section, the authors mentioned a few applications for tracking a user’s location such as car navigation, fitness tracking and emergency services. However, there are many other applications for location tracking technologies including vehicle activity recognition [1], point-of-interest recommendation [2], and Geomarketing [3]. Please review the following papers as a good starting point and include the appropriate references for each of these applications.
[1] 10.1109/TITS.2020.2970229
[2] 10.1109/TKDE.2020.3007194
[3] https://doi.org/10.3846/16111699.2015.1113198
2) In the related works section, the list of BLE-based applications highlighted by the authors is rather short and lacking other popular BLE applications, such as BLE-based occupancy prediction [1], BLE-based emergency management [2], and BLE-based smart energy management system [3]. Please review these established applications in the Background section to highlight the importance of BLE for a wide range of applications.
[1] https://doi.org/10.1016/j.buildenv.2020.106681
[2] https://doi.org/10.1007/978-3-319-47217-1_25
[3] https://doi.org/10.1016/j.buildenv.2022.109472
3) Aside from reviewing the existing literature, it is useful to clearly highlight and discuss the limitations or gaps in these studies that will be addressed in this manuscript.
4) During the calibration process, the authors mentioned that the users are asked to stand in the middle of each region and remain stationary for a period of 180 seconds. However, this behaviour does not reflect an individual’s actual behaviour, which can include sitting, lying and walking around in the room. How would these behaviours have an impact on the model’s performance?
5) In Figure 8, most of the results make sense as the points closer to the beacons and away from the region boundaries are showing higher accuracies than the points that are close to the boundaries of each region. However, there are two points do not follow this trend (i.e., P3 for Region 0 with a low accuracy of 73.3% when it is next to Beacon 0 and P2 in Region 1 with a high accuracy of 100% even when it is close to Region 0). Please kindly provide an explanation for these results.
6) Please include a discussion section on the limitations of the existing approach and elaborate on how they can be improved in future works. Some discussion points include:
· Depending on the adoption rate in different regions, it might not be very common that every individual has a smartwatch. In this scenario, does this approach work with the individual’s smartphone?
· Given that there is a need to recalibrate and collect the RSSI values at the reference points every time the system is deployed in a new location/building, this will have a negative impact to the approach’s scalability. How will this be resolved in the future?
7) Minor comments:
- There is a formatting issue in the authors’ email addresses, where extra superscriptions containing each authors email (i.e., 4,5,6). Also, please only keep the corresponding authors’ email, the rest can be removed to avoid information crowd.
- Missing “,” in line 31 after “year 2050”.
- Suggest to rename the “Background” section to “Literature Review” or “Related Works” as it is a more common naming convention.
- In Figure 4, please also include the location where the individual was standing when he/she is performing calibration.

Comments on the Quality of English LanguageThere are only some minor issues with the quality of English language, which have been highlighted in my earlier comments.
Reviewer 3 Report
Comments and Suggestions for Authors
First of all, I want to congratulate the authors for their efforts in this manuscript. The topic that they present, even though it is not novel, is interesting and fits within the journal's scope. They have analyzed the use of Bluetooth Beacons and RSS Fingerprinting with Smartwatch for indoor locations. In general terms, the paper is well-structured and provides accurate results. There are a few issues to be solved before accepting the paper:
Abstract and keywords:
The abstract should be extended to reach 250 words approx.
In the abstract, the authors have to highlight their results, including numerical values of the performance of their proposal.
Avoid using as a keyword, terms already used in the title. Deleted keywords included in the title and provided new keywords.
Introduction:
There is a lack of references in the introduction to contextualize certain issues. Please provide references to justify the information included in the introduction.
Some paragraphs of the introduction are extremely short. The authors have to extend the provided information in short paragraphs or consider merging them.
Authors should explain the structure of the rest of the paper at the end of the introduction section.
Background:
I find missing the following reference in the background section:
SR Jondhale, V Mohan, BB Sharma, J Lloret, SV Athawale, Support vector regression for mobile target localization in indoor environments, Sensors 22 (1), 358. 2022
Methods:
Please provide the datasheet as a reference for used equipment/devices or include the manufacturer's name and location in the main text.
Figure 1 should include a scalebar or the size of the studied region.
Evaluation
Same comment in Figure 1 for Figures 3 and 4.
Results:
I suggest adding in Table 3 the results of an ANOVA procedure in order to evaluate if the differences observed are significant or not.
Same comment in Figure 1 for Figure 5.
Conclusion:
Future work is missing at the end of the conclusion section.